# Retrospective Analysis of the Ventilatory Workload Kinetic Index during Stability and Crisis in Patients with Asthma and COPD in a Monitored Program

**DOI:** 10.3390/geriatrics9020029

**Published:** 2024-03-01

**Authors:** Rodrigo Muñoz-Cofré, Ramón Pinochet-Urzúa, Mariano del Sol, Paul Medina-González, Jorge Valenzuela-Vásquez, Gerardo Molina-Vergara, Rodrigo Lizama-Pérez, Máximo Escobar-Cabello

**Affiliations:** 1PhD Program in Morphological Sciences, Universidad de La Frontera, Temuco 4811230, Chile; rodrigomunozcofre@gmail.com (R.M.-C.); mariano.delsol@ufrontera.cl (M.d.S.); 2Hospital Padre Alberto Hurtado, San Ramón 8880465, Chile; rpinochetu@gmail.com; 3Center of Excellence in Morphological and Surgical Studies, Universidad de La Frontera, Temuco 4811230, Chile; 4Department de Kinesiology, Universidad Católica del Maule, Talca 3480112, Chile; paulmedinagonzalez@gmail.com; 5Physical Medicine and Rehabilitation Service, Hospital el Carmen, Maipú 9251521, Chile; jorge.valenzuelav@redsalud.gob.cl; 6Department of Education of the Illustrious, Municipalidad de Quirihue, Quirihue 4000000, Chile; gamolinavergara@gmail.com; 7Department of Physical Education and Sport, Faculty of Sport Science, University of Granada, 18071 Granada, Spain; rlizamaprez@gmail.com

**Keywords:** ventilatory assessment, physiotherapy, asthma, chronic obstructive pulmonary disease

## Abstract

To assess ventilatory evolution through the Ventilatory Workload Kinetic Index (VWKI) in patients with asthma and chronic obstructive pulmonary disease (COPD) during stability and exacerbation. Retrospective analysis. Conducted at the Padre Alberto Hurtado Hospital, Santiago, Chile. Ten patients with asthma and fifty-five with COPD participated. Sixty-five clinical records were reviewed. The VWKI in stability and exacerbation of these patients was extracted. When analyzing the baseline with the peak in both asthma and COPD, there was a significant increase in the VWKI. Similarly, the loads, translations, and supports significantly increased from the baseline to the peak. However, in the loads, there were no changes in airway resistance for asthma or in cough for COPD. Likewise, the supports for asthma and COPD showed no changes in the O_2_. The VWKI determined ventilatory issues in outpatients and made locating the greatest compromise in loads, translations, or supports possible.

## 1. Introduction

Physical therapy is a field of professional action aimed at solving health problems linked to human movement dysfunctions. Traditionally, these complications have been resolved by consulting a Kinesiology Vademecum; however, this represents a fundamental complement to the therapy [1]. In this context, a set of scores is utilized to assess the severity of a respiratory condition. These values are assigned to provide guidance for clinical interventions and decision-making processes [2].

Due to their ability to provide a sense of orientation, these scales have gained great significance for children and adults with acute respiratory diseases [2]. In addition, the results help measure the effectiveness of the interventions, allowing for the possibility of reworking treatment strategies [3].

In this context and to have a correlation between evaluation and therapeutic action, as well as to objectify the different profiles of ventilatory dysfunction, the Ventilatory Workload Kinetic Index (VWKI) was proposed as a clinical assessment tool to analyze the ventilatory balance-imbalance and to standardize the evaluation of the respiratory system by the physiotherapist [4,5]. The VWKI is a clinical instrument with high inter-evaluative reliability (*p* = 0.9, K = 0.84). It comprises eight variables, which, in their all-around clinometric capacity, have proven to be a good differentiator of functional contexts related to respiratory problems in hospitalized patients [6,7], including the patients of nocturnal physiotherapy clinics [5,8].

Asthma is understood as a variable disease normally characterized by chronic airway inflammation. Respiratory symptoms are wheezing, difficulty breathing, chest tightness, and cough, which vary over time and in intensity, and with varying expiratory airflow restriction [9]. Chronic obstructive pulmonary disease (COPD), a preventable and treatable disease, is characterized by dyspnea, cough, and/or production of sputum, as well as a persistent restriction of airflow due to anomalies of the respiratory and alveolar tracts caused by particles or noxious gases [10].

Asthma and COPD are highly prevalent worldwide; in Chile, they are 9 and 16.9%, respectively [11,12]. During the natural course of asthma and COPD, there are exacerbating events called crises in the case of asthma, where there are whistling sounds, respiratory difficulties, chest tightness, and cough [9]. Exacerbations in COPD are characterized by dyspnea, coughing, the production of sputum, and persistent airflow limitation. This causes lung ventilation deterioration, directly impacting the affected patient’s quality of life [10]. In addition, it is a significant public health concern due to the high mortality rate among elderly adults and the high cost of treating its advanced stages for outpatients and inpatients [13].

Considering this, in 2001, the Chilean Ministry of Health created the Adult Respiratory Diseases (ARD) program [13]. This includes a monitoring plan, the main objective of which is to reduce morbidity and mortality in patients. This was a challenging scenario for the general health team and the physiotherapy team in particular, where evaluating the condition of this population and observing their evolution was a cornerstone for successful treatment [12,13].

This study aimed to compare the VWKI variables, in stability and crisis/exacerbation, in patients diagnosed with asthma and COPD in the ARD program.

## 2. Materials and Methods

The study design was retrospective. The VWKI was obtained from the patients monitored in the ARD program at Padre Alberto Hurtado Hospital, Santiago, Chile, between March and September 2017. The current regulations of this program state that patients with severe chronic respiratory diseases will be monitored in secondary health facilities (referral hospital). For these patients, the check-ups will be every 3 months with a bronchopulmonary specialist and monthly with a physiotherapist. The VWKI was compared in terms of stability and accuracy.

Patients with a stable history for the six months before the date of deterioration were included. From the 65 clinical records, the following information was obtained: (i) medical diagnosis and the post-bronchodilator spirometric values of the last control spirometry, (ii) the VWKI baseline corresponded to the median of the last six months’ assessments of stable controls, and (iii) the KIVW in exacerbation or crisis was taken from attendance outside the scheduled check-up and with symptomatology consistent with the definitions of GINA [9] for asthma and GOLD for COPD [10]. To guarantee the validity of these results, only the records of the resident physiotherapist (RMC), a specialist in cardiopulmonary rehabilitation, were considered, and the bronchopulmonary specialist in charge made the diagnosis of exacerbation or crisis. The Scientific Ethics Committee at Maule Catholic University (resolution 23/2016) and the Physical Therapy Coordinator at the Padre Alberto Hurtado Hospital approved the study. 

### 2.1. Classification and Categorization of the VWKI

The VWKI is a clinical instrument consisting of eight variables, each with a score ranging from zero to three points according to clinical commitment (Table 1). These are classified as loads, internal or external biophysical phenomena that increase the mechanical or physiological expenditures of the ventilatory system; translations, a set of variables for adequate monitoring of the tendencies towards imbalance in the system; and supports, internal or external biophysical adjustments that stabilize the equilibrium costs of the ventilatory system at a given moment [4]. Detail of the measured variables (Figure 1A):

#### 2.1.1. Respiratory Rate (RR)

The number of breaths per minute was measured with a Casio chronometer (model Hs-3v-1b).

#### 2.1.2. Additional Oxygen (O_2_)

The additional oxygen support administered was measured as a percentage, regardless of the system used (high or low flow).

#### 2.1.3. Oxygen Saturation (SO_2_)

Was recorded with an oximeter with a NONIN pulse (ONYX 9500) attached to the index finger of each patient.

#### 2.1.4. Use of Accessory Muscles (UAM)

Accessory muscle activity was measured by observation and/or contact in a sitting position and with minimum intervention to gauge the level more clearly.

#### 2.1.5. Pulmonary Murmur (PM)

This was measured with a 3M™ Littmann^®^ Classic III stethoscope (Saint Paul, MN, USA). The central points of each of the ten quadrants were estimated at total lung capacity. Specifically, the front quadrants were located at the two apices, the side quadrants were positioned at the two bases, and the back quadrants were identified at the two higher points, two middle points, and two lower points. Each location was awarded points: 0 points, restrained pulmonary murmur, 1 point, diminished pulmonary murmur, and 2 points suppressed pulmonary murmur. The sum of the ten locations was categorized according to Table 1.

#### 2.1.6. Airway Resistance (AR)

Once the inspiratory and expiratory phase had been delimited, the presence or absence of prolonged expiration, expiratory wheezing, or biphasic wheezing was auscultated with a 3M ™ Littmann^®^ Classic III stethoscope (Saint Paul, MN, USA). The data were scored according to Table 1.

#### 2.1.7. Cough (Cough)

Its evaluation was clinical, determined by the kinematic observation of a voluntary coughing effort: (i) normal presence of the three phases, (ii) upset to trigger or preparation stage (volume of inspiratory reserve), alteration of the compressive expulsive phase, and absent mechanism.

#### 2.1.8. Attempts to Permeabilize the Airway (APA)

The number of times needed to repeat the procedure was established so the physiotherapist could check if the airway was cleared.

The sum of the eight variables yields a total VWKI score categorized as mild, moderate, or severe ventilatory compromise [5] (Figure 1B).

### 2.2. Statistical Analysis

To analyze the results, Microsoft Office Excel^®^ 2010 was used to tabulate the data, and GraphPad Prism 5^®^ was used for the statistical analysis. The data were presented with median and interquartile ranges and/or mean ± standard deviation. The analysis began by establishing the normality of the data via the Shapiro–Wilk test. According to the data distribution, a Student’s test or Mann–Whitney U test was used.

Finally, a level of significance of *p* < 0.05 was considered to determine the existing variability in the intra-pathological and inter-pathological ventilatory overload. 

## 3. Results

Of the 65 patients, 10 had asthma and 55 COPD. They were divided as follows: asthma 6 women and 4 men, COPD 31 women and 24 men. The categorization was severe for both asthma and COPD (Table 2). 

The loads, translations, and supports underwent a significant increase from the baseline to the peak (Table 3). However, in the impediments, there were no changes in AR for asthma or in cough for COPD. Similarly, in the supports for asthma and COPD, there were no modifications in the O_2_ (Figure 2A,B). When comparing the baseline state and the peak, there was a significant increase in the VWKI in both asthma and COPD, from 8 to 14 and 9 to 15 points, respectively (Figure 2C,D). It is worth noting that despite this change, the categorization remained the same for moderate ventilatory compromise (Figure 2C,D).

## 4. Discussion

The aim of this investigation was to compare the variables of VWKI in patients with asthma and COPD belonging to the ARD program, during stability and exacerbation. The results of this study make it possible to differentiate the baseline from the peak of the crisis/exacerbation. In this context, the use of the VWKI enables visualization changes in the variables that comprise it, as well as detecting changes in the loads, translations, and supports that the system incorporates and their impact on the total score of the VWKI (Figure 3).

It is important to highlight that this clinometric tool is widely used because the measurement of its parameters does not require instrumental sophistication. This is complemented by Roberts et al. (2018), who indicate that members of the health team in training need to use specialized skills and tools in this area to improve the diagnosis and treatment of the respiratory disorders mentioned above [3,14]. Thus, the systemization of the use provided by the VWKI would allow for the stratification of the risk of ventilatory failure in hospitalized patients and objectively and specifically address the respiratory physiotherapeutic treatment of each patient according to their workloads [5]. In this context, the information provided by the VWKI can contribute to decision making in the search for a more efficient use of economic and health resources, both within and beyond the hospital environment [12,13,14].

In the present study, despite having an unbalanced sample by type of pathology and number of participants in the program, this characteristic is consistent with Chile’s sociodemographic reality [11,12]. However, depending on the pathophysiology of the patients evaluated, they presented a partial deterioration, since as a percentage they were higher in asthma than in COPD, and presented less change, 16% vs. 25%, respectively (Figure 3, Table 3). In both asthma and COPD, there was a significant difference in these variables between their initial behavior and in crisis/exacerbation. 

The discriminatory capacity of the VWKI agrees with that reported by Pinochet et al. (2004), who evaluated patients with noninvasive mechanical ventilation, finding a value that ranged between two and three points for all variables on the VWKI [5]. The implications for physiotherapy are established from the specificity of the patient evaluated, since the different VWKI profiles translate to different pathophysiological conditions; so, we consider it important to analyze and present the total score in the VWKI in their respective loads, translations, and supports (Figure 2A,B). In this sense, Cancino et al. (2004) studied the behavior of the VWKI during the night shift at the Padre Hurtado Hospital, where 291 evaluations were performed in 64 patients, observing that 81% obtained between 9 and 16 points, and 4% exceeded 16 points. The authors concluded that 85% of the patients had moderate to severe ventilatory compromise [8].

To contribute to solving this problem, Quintero et al. (2014) attempted to describe and disseminate the role of the VWKI in the intervention of the hospitalized patient. Given the deficient application of a correct model of physiotherapeutic care for patients with ventilatory impairment, combined with the lack of specialist training in this area, they assert that this has hampered the health care process in the treatment of respiratory pathologies [15]. They suggest that it is vitally important to promote the dissemination of tools that can complement a correct examination, evaluation, diagnosis, prognosis, and treatment of patients who require respiratory physiotherapy [16]. Thus, systematizing the use that the VWKI provides would permit risk stratification of ventilatory failure in hospitalized patients and objectively and specifically address the respiratory physiotherapy of each patient according to their workloads. Hence, a consideration of compromise made on the basis of the total ventilatory workload increase score alone will not necessarily involve identical altered variables with similar scores (Figure 2A,B).

### 4.1. Effects of the Asthma Crisis on the VWKI

In the case of loads, an asthma crisis causes a reduction in PM compared to the baseline. The GINA indicates that this reduction occurs mainly if there is a severe obstruction, culminating, in some extreme cases, in a silent chest, characterized by an absence of audible wheezing [9]. 

By contrast, increased AR is characteristic in an asthma crisis, where a prolonged expiration or expiratory wheezing can be heard even in the inspiratory phase [9,17]. This could be due to a dysfunction of the smooth muscles of the airway, which present a change in the structure and mechanical properties of the contractile components due to acute or chronic inflammation of the respiratory tract [18]. Both situations are reflected in the score assigned on the VWKI: the PM increased significantly and the AR (Figure 3B) kept its high score (Table 3). 

According to Silva et al. (2013), asthmatic patients may require the UAM to facilitate and even allow ventilation, so they would change from being accessory muscles to being the main muscles of inspiration when the diaphragm does not have a mechanical advantage [19]. This is reflected in the score of these variables on the VWKI during a crisis, where the scores for RR and UAM increased significantly, but not so much SO_2_ (Table 3). With regard to the APA, the literature indicates that during an asthma crisis, there is difficulty in self-permeabilizing the airway due to changes in the rheology of the mucus, increasing its viscoelasticity and adhesiveness due to dehydration and making its transport and subsequent elimination difficult [20]. This is clearly noted in the score obtained by this variable on the VWKI (Table 3).

### 4.2. Effects of COPD Exacerbation on VWKI

The loads in COPD had no changes except for the cough, which generated a lower change percentage between the baseline and exacerbation (Figure 3C,D). In this respect, Vanfleteren et al. (2016) state that air trapping, produced by the difficulty the alveoli have in emptying their air content, is one of the many causes that leads to hyperinflation [21]. This is caused by the loss of elastic retraction in the lungs, increasing the functional residual capacity (FRC) when at rest or the end-expiratory lung volume (EELV) of an above average expiration [22]. This occurs because of the loss of elastic recoil of the lungs, increasing the TLC by increasing the RR. 

Also, in terms of AR, Ha and Rogers (2016) indicate that the rise in mucin production and its increased exocytosis in the secretory cells of the respiratory tract increases the density and viscosity of the mucus gel located on the surface of the epithelium, reducing the bronchial lumen [20]. Both variables underwent a significant increase in their score; therefore, their behavior aligned with previously discussed points (Table 3).

In the same way, the APA increased from one to two points. In this regard, Ha and Rogers (2016) reported that the rheological changes to the mucus, the structural changes in the airway and the alteration in the cough mechanism can induce changes in the clearance of the airway [20,23]. This ultimately makes it difficult to eliminate the secretions expressed in the 3-point categorization on the VWKI at peak (Table 3).

The limitations of this study include the number of patients with asthma being small, which would be related to the prevalence ratio in relation to COPD (9 and 16.9%, respectively). As the VWKI is a clinical tool, it requires experience and initial training to be able to characterize subjective variables, such as PM or cough [4,16]. Thus, we plan for future research to observe inter-rater reliability with this tool and also to implement a monitoring of chronic respiratory patients with the VWKI that allows patient education in relation to their periods of stability and the recognition of atypical symptoms that will allow them to consult their treating physician in time.

## 5. Conclusions

The VWKI serves to identify ventilatory issues in outpatients and can also distinguish where the greatest impairment lies in terms of loads, translations, or supports. In this context, it may be considered a useful professional tool since it can support referral, allowing a more efficient use of human and economic resources of programs, and improving the quality of care for chronic respiratory patients by inducing a more consistent respiratory physiotherapy service. 

## Figures and Tables

**Figure 1 geriatrics-09-00029-f001:**
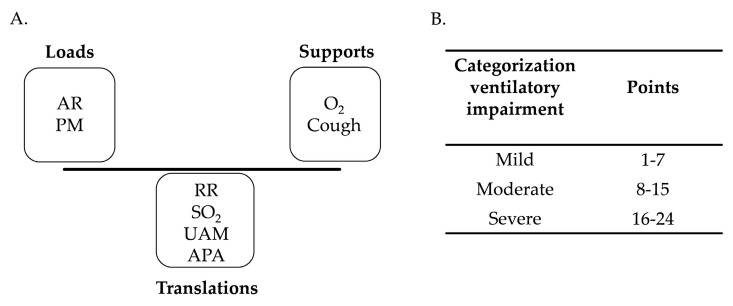
(**A**). classification of loads, translations, and supports. (**B**). categorization of total VWKI score. RR: respiratory rate; O_2_: additional oxygen contribution; SO_2_: oxygen saturation; UAM: use of accessory muscles; PM: pulmonary murmur; AR: airway resistance; APA: attempts to permeabilize the airway.

**Figure 2 geriatrics-09-00029-f002:**
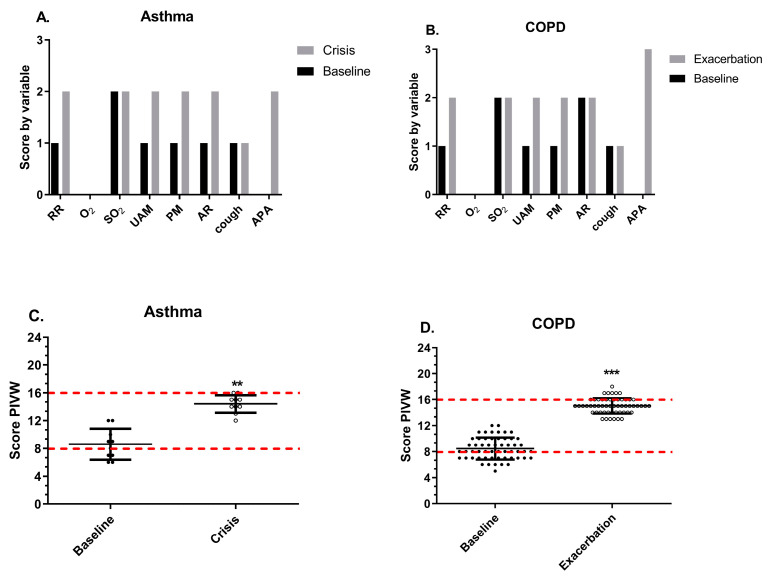
Comparison of median scores in asthma and COPD. (**A**) VWKI score by its variables in asthma comparing the baseline and crisis. (**B**) VWKI score by its variables in COPD comparing the baseline and exacerbation. (**C**) Total VWKI score in asthma comparing the baseline and crisis. (**D**) Total VWKI score by its variables in COPD comparing the baseline and exacerbation. In (**C**,**D**), the red line represents the change in the categorization of the ventilatory pattern. RR: respiratory rate; O_2_: additional oxygen; SO_2_: oxygen saturation; UAM: use of accessory muscles; PM: pulmonary murmur; AR: airway resistance; APA: attempts to permeabilize the airway. **: *p* < 0.01; ***: *p* < 0.001.

**Figure 3 geriatrics-09-00029-f003:**
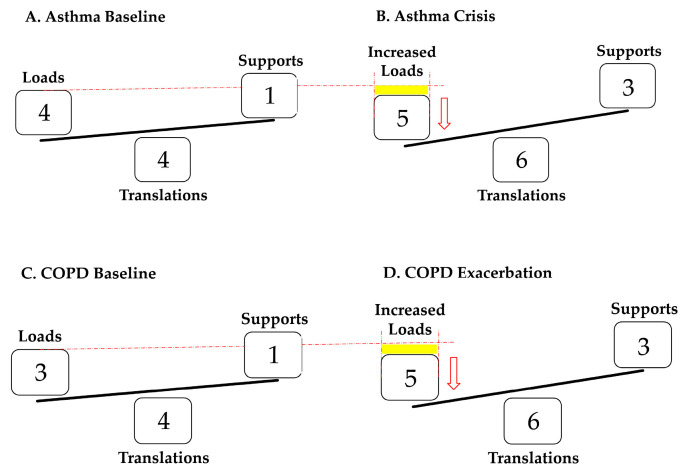
(**A**,**B**) shows the behavior of patients with asthma at baseline and in crisis; an increase in the loads in the crisis is observed, shown by the yellow rectangle, which causes a negative imbalance that represents the reduction in PM since AR is maintained. (**C**,**D**) shows the behavior of patients with COPD at baseline and in exacerbation, also showing an increase in loads in exacerbation. However, the difference is that, in such cases, they correspond to the joint increase in AR and PM. Both translations and supports are increased without distinction between the two pathologies.

**Table 1 geriatrics-09-00029-t001:** Division and score of the variables of the of the Ventilatory Workload Kinetic Index.

RR	O_2_ (%)	SO_2_ (%)	UAM	PM	AR	Cough	APA	SCORE
10–16	100–98	21	Without UAM	0	Without AR	Spontaneous or effective cough	Not required	0
17–25	97–95	22–28	Diaphragmatic overload	1–7	Prolonged expiration	Threshold disorder or inspiratory reserve volume ↓	2 attempts	1
26–34	94–92	29–49	AMR I or E	8–14	Wheezing or expiratory rhonchi	Compressive or expulsive phase altered	3–4 attempts	2
35+	<91	>50	AMR I and E/ PR	15–20	Wheezing or expiratory and inspiratory rhonchi	Absent or severely altered mechanism	>5 attempts	3

RR: respiratory rate; O_2_: additional oxygen contribution; SO_2_: oxygen saturation; UAM: use of accessory muscles; PM: pulmonary murmur; AR: airway resistance; APA: attempts to permeabilize the airway; AMR: accessory muscle recruitment; I: inspiratory; E: expiratory; PR: paradoxical respiration. Adapted with permission from Escobar et al., 2000, copyright owner’s Escobar Cabello. ↓: decreased inspiratory reserve volume.

**Table 2 geriatrics-09-00029-t002:** Characterization of the study group.

	Asthma (*n* = 10)		COPD (*n* = 55)	
VARIABLE	Female	Male	Female	Male
Number and percentage	6 (60%)	4 (40%)	31 (56.36%)	24 (43.64%)
Age (years)	62.16 ± 7.90	67.66 ± 0.47	63.20 ± 8.67	66.16 ± 7.10
Weight (kilograms)	78.83 ± 13.77	77.00 ± 9.79	68.68 ± 21.58	65.29 ± 14.19
Height (centimeters)	145.83 ± 2.78	159.00 ± 6.12	152.37 ± 6.95	163.20 ± 4.22
FVC (% of prediction)	72.16 ± 9.72	55.66 ± 14.61	69.58 ± 11.44	59.29 ± 11.64
FEV_1_ (% of prediction)	52.50 ± 14.88	29.66 ± 10.37	46.48 ± 7.59	35.87 ± 10.60
FEV_1_/FVC (%)	48 ± 2	34 ± 3	41 ± 2	31 ± 1

FVC: forced vital capacity; FEV_1_: forced expiratory volume in 1 s. The measurements for the female and male are reported as mean ± standard deviation. Post-bronchodilator spirometry.

**Table 3 geriatrics-09-00029-t003:** Statistical significance of the loads, translations, and supports.

	Asthma			COPD		
	B	C	*p* Value	B	E	*p* Value
Loads	4	5	0.005	3	5	0.0001 ^£^
PM	1 (1–2)	2 (1–3)	0.04	1 (1–2)	2 (1–3)	0.0001 ^#^
AR	2 (1–3)	2 (2–3)	1.000	1 (2–3)	2 (2–3)	0.0015 ^#^
Translations	4	6	0.005	4	6	0.0001 ^£^
RR	1 (1–3)	2 (2–3)	0.009	1 (1–2)	2 (1–3)	0.0001 ^#^
SO_2_	2 (2–3)	2 (1–3)	0.900	2 (0–3)	2 (1–3)	0.998 ^£^
UAM	1 (1–2)	2 (2–3)	0.005	1 (1–2)	2 (0–3)	0.0001 ^£^
APA	1 (1–3)	3 (1–3)	0.008	1 (1–2)	3 (2–3)	0.0001 ^#^
Supports	1	3	0.008	1	3	0.0001 ^#^
ApaO_2_	0 (0–0)	0 (0–1)	1.00	0 (0–1)	0 (0–2)	0.11 ^£^
Cough	1 (1–2)	1 (1–2)	0.102	1 (1–2)	1 (1–2)	0.100 ^#^

The data are presented in median (minimum-maximum). B: baseline; C: crisis; E: exacerbation; ^#^: Mann–Whitney U; ^£^: t-student; RR: respiratory rate; ApaO_2_: additional oxygen; SO_2_: oxygen saturation; UAM: use of accessory muscles; PM: pulmonary murmur; AR: airway resistance; APA: attempts to permeabilize the airway.

## Data Availability

The datasets generated and/or analyzed during the current study are available from the corresponding author upon reasonable request.

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
