# Peer review of "Retrospective Analysis of the Ventilatory Workload Kinetic Index during Stability and Crisis in Patients with Asthma and COPD in a Monitored Program"

_geriatrics, 2024, doi:10.3390/geriatrics9020029_

Round 1

Reviewer 1 Report

The subject could be interesting, but not really useful for clinicians.

The data were collected in exacerbations? because is not clear. if the patient answer that they are stable in the last 6 months, it is possible to forget some exacerbation (mild exacerbation for example).

How were recruited patients? the seasonality of the symptoms, the variability of the symptoms.

Also, the number of asthmatic patients is too low ( only 10 ?) couldn't be relevant...also, the patients with asthma had very low values of spirometry test (less than COPD?). It is possible to have a bias in the selection of the study population.

patients with asthma are controlled?

In severe exacerbations with hospitalisations physiotherapy and rehab are different than in stable conditions. So, I am not sure that the hypothesis and the results are correct, and sure are not clearly exposed.

Author Response

Estimate Reviewer
Attached PDF with explanatory table of the changes made in the document

Reviewer 2 Report

Dear authors 

I have some comments requiring discussion 

1) please revise language. Some paragraphs lack of clarity.

2) figure 3 is not clear

it is not clear how it has been built and it’s significance. I think that caption and linked text should be reformulated. Otherwise it remains too unclear.

3) I cannot understand how an exacerbated COPD or asthma patient cannot have modifications of AR! This is the basis of an exacerbation or asthma attack. In addition, it is well known that COPD exacerbation are characterised by cough impairment and sputum volume augmentation. This is also in contrast with the pulmonary function test shown in the manuscript (see tiffaneau index!)

4) there is a large discrepancy between samples is asthma and copd patients

this is a major problem in the manuscript 

5) it is not defined how authors define severity or not in both populations. please indicate criteria.

6)discussion would benefit of a revision in its form. Now it sounds more like a review than a discussion of findings. Although the contents of discussion are ok, please try to reformulate it according to my suggestion 

Check all the manuscript 

Author Response

(The authors gave the same response as above.)

Round 2

Reviewer 1 Report

Could be publish in recent form.

Reviewer 2 Report

no further comments